# Explore In-Context Learning for
# 3D Point Cloud Understanding

**Zhongbin Fang**[1], **Xiangtai Li**[2], **Xia Li**[3], **Joachim M. Buhmann**[3], **Chen Change Loy**[2]
**Mengyuan Liu**[4] ✉

[1]Sun Yat-sen University  [2]S-Lab, Nanyang Technological University
[3]Department of Computer Science, ETH Zurich
[4]Key Laboratory of Machine Perception, Shenzhen Graduate School, Peking University
fangzhb5@mail2.sysu.edu.cn, xiangtai.li@ntu.edu.sg, xia.li@inf.ethz.ch
ccloy@ntu.edu.sg, liumengyuan@pku.edu.cn
https://github.com/fanglaosi/Point-In-Context

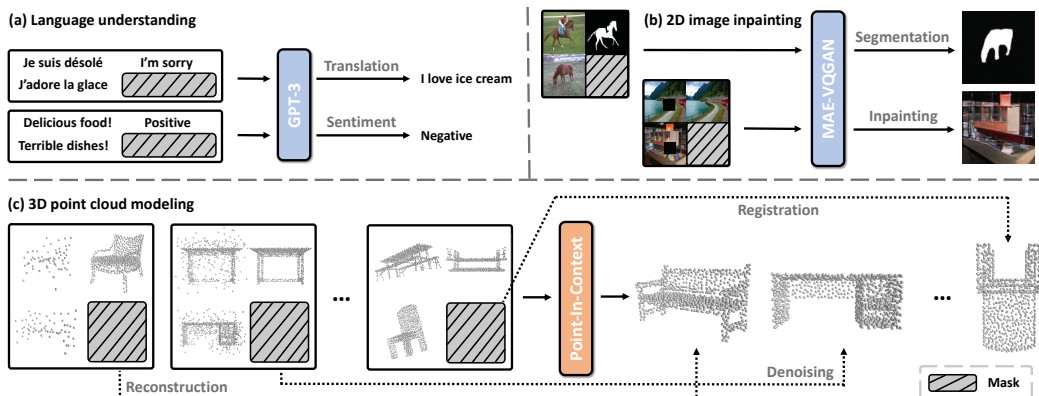

Figure 1: (a) In-context learning in NLP [6], with different text prompts for corresponding tasks: translation and sentiment analysis. (b) In-context learning in 2D vision [4], with 2D visual prompts for different tasks: segmentation and inpainting. (c) Our proposed in-context learning for 3D point clouds, with 3D visual prompts for different tasks: reconstruction, denoising, registration, etc.

## Abstract

With the rise of large-scale models trained on broad data, in-context learning has become a new learning paradigm that has demonstrated significant potential in natural language processing and computer vision tasks. Meanwhile, in-context learning is still largely unexplored in the 3D point cloud domain. Although masked modeling has been successfully applied for in-context learning in 2D vision, directly extending it to 3D point clouds remains a formidable challenge. In the case of point clouds, the tokens themselves are the point cloud positions (coordinates) that are masked during inference. Moreover, position embedding in previous works may inadvertently introduce information leakage. To address these challenges, we introduce a novel framework, named Point-In-Context, designed especially for in-context learning in 3D point clouds, where both inputs and outputs are modeled as coordinates for each task. Additionally, we propose the Joint Sampling module, carefully designed to work in tandem with the general point sampling operator, effectively resolving the aforementioned technical issues. We conduct extensive experiments to validate the versatility and adaptability of our proposed methods in handling a wide range of tasks.

---

✉ Corresponding author is Mengyuan Liu.

37th Conference on Neural Information Processing Systems (NeurIPS 2023).

# 1 Introduction

In recent years, large-scale models [1, 32, 15, 5] with enormous parameters pre-trained on broad data have emerged in computer vision and natural language processing. Capable of handling diverse tasks simultaneously, these models can be adapted for new tasks when prompted. Text-to-image generation models such as DALL-E [34] and language models like GPT [33] are examples of this development, representing significant progress toward general intelligence. However, training these models is resource-intensive, which makes full fine-tuning [18, 21, 20] or even parameter-efficient tuning techniques [50, 17, 25] such as prompt tuning impractical for many users.

In-context learning, originating from natural language processing (NLP) [33, 35, 6], holds potential as the mainstream approach for efficient model adaptation and generalization. Unlike other methods that necessitate model parameter updates for new tasks, in-context learning incorporates domain-specific input-output pairs, known as in-context examples or prompts, into a test example. In NLP, the prompt can be a machine translation pair or sentiment analysis, as shown in Fig. 1 (a). This allows the model to produce optimal results without requiring any parameter updates for previous tasks. Several works [4, 38, 39, 48] explore in-context learning in computer vision. The visual prompt [4] is the first to adopt a pre-trained neural network for filling missing patches in grid-like images, as shown in Fig. 1(b). Other studies investigate the effect of visual prompts or generalization to more vision tasks. The above methods adopt Mask Image Modeling (MIM) architecture [13, 3] for in-context task transfer. Inspired by MIM, Masked Point Modeling (MPM) is proposed recently and is widely used in different point cloud tasks [44]. To our knowledge, no work has explored in-context learning for 3D point cloud understanding using the MPM framework.

Our work is the first to explore in-context learning for 3D point cloud understanding, as shown in Fig. 1(c). Given the absence of previous works, we propose a benchmark based on the ShapeNet [7] and ShapeNetPart [42], encompassing four different tasks: point cloud reconstruction, denoising, registration, and part segmentation. Meanwhile, we benchmark several representative baselines [27, 40, 26, 12], including individual models for each task and the models equipped with shared backbone with multiple task heads. Then, to tackle the benchmark mentioned above, we present the Point-In-Context (PIC) to explore in-context learning for 3D point clouds.

A straightforward extension of the 2D MIM architecture to point clouds for in-context learning may encounter two primary obstacles. First, using the conventional position embedding approach could potentially lead to information leakage, attributed to the use of invisible center points[1]. Second, unlike 1D word embeddings or 2D images, 3D point cloud data, which are inherently unordered [27], present the risk of having their positional information become disarrayed when partitioned into a patch sequence. Consequently, it is indispensable to devise a new sampling and grouping strategy for 3D point cloud data. In response to this issue, we propose a simple yet effective solution, termed joint sampling. This technique involves recording the indices of sampled center points and using the K-nearest neighbor strategy to sample both the input and target concurrently. In addition, leveraging the mask point transformer architecture, we explore two distinct baseline methodologies for PIC, which encompass separating inputs and targets akin to the Painter strategy [38] and concatenating inputs and targets in a manner analogous to the MAE approach [13] for reconstruction. Contrary to the focus of few-shot learning on a single specific task, our objective is to explore the in-context ability of the generative model, facilitating the execution of tasks commensurate with the 3D prompt.

Our main contributions are as follows. 1) We introduce a simple yet effective general framework for 3D visual prompting. Given two pairs of point clouds, we demonstrate that multiple 3D point cloud tasks can be treated as task-aware prompting, as seen in 2D image and NLP tasks. 2) We create a new benchmark, including four different point cloud tasks for 3D in-context learning, and evaluate several representative baselines. 3) Our comprehensive study addresses various 3D prompt examples, sampling methods, and masking strategies. Moreover, we show that the choice of in-context examples greatly impacts performance for 3D in-context learning.

# 2 Related Work

**3D Point Cloud Classification.** Deep neural networks have been proposed for the 3D point cloud classification task. Both PointNet [27] and its improved versions [28, 30] are pioneers of point-based

---

[1]Previous MPM methods [44, 26] embed position with the masked center points during pre-training.

methods in 3D point cloud analysis, which adopt multi-layer perceptron (MLP) to handle point clouds directly. Several graph-based methods [40, 22, 16] exploit geometric properties and propose different dynamic kernels. In particular, DGCNN [40] designs EdgeConv, which dynamically computes each layer output. Recently, several works [49] adopt pure transformer-based architecture to model the global context. Point Transformer [49, 12, 43] applies the vectorized self-attention mechanism to construct a point Transformer layer for 3D point cloud learning. Meanwhile, PointMLP [24] directly applies a pure residual MLP network to the 3D point cloud analysis. Recently, several works have explored the vision language models [46] and joint 2D and 3D training [31, 47] via transformer architectures. However, these approaches are only designed for a single task, so they cannot be used directly for in-context learning.

**Masked Image Modeling (MIM) For 2D and 3D Vision.** The GPT and BERT series [8] have greatly enhanced natural language processing performance through masked modeling and fine-tuning downstream tasks. BEiT [3] is the first to propose matching image patches with discrete tokens via d-VAE [34] and pre-train a standard vision transformer [37, 10] using masked image modeling [8]. MAE [13] then directly reconstructs the raw pixel values of masked tokens and achieves high efficiency with a high mask ratio. For 3D point cloud pre-training using MIM, several works [44, 26, 45, 47, 9, 29, 19] aim to improve feature representation. Point-BERT [44] adopts a BERT-like architecture, while Point-MAE [26] transfers the MAE-like framework for pre-training. These methods use standard transformer networks to process 3D point clouds and achieve competitive performance on various downstream tasks. Our approach follows a similar point MIM pipeline but explores the in-context ability of point transformers and MIM, which has not been investigated previously.

**In-Context Learning.** In-context learning [33, 35, 6] is a new learning paradigm in large language models like GPT-3. This paradigm enables an autoregressive language model to perform inference on unseen tasks by conditioning the input on specific input-output pairs, known as "context." This powerful paradigm allows users to customize a model's output to fit their downstream datasets without modifying the often inaccessible internal model parameters. Recent research in natural language processing has demonstrated the efficacy of in-context learning across a range of language tasks, including machine translation, sentiment analysis, and question-answering. Recently, several works [4, 38, 39, 48, 36, 2] explore in-context learning in computer vision. The visual prompt [4] is the first pure vision model, which was pre-trained to fill missing patches in images that are made of academic figures and infographics. Then, Painter [38] generalizes visual in-context learning as learning to paint the image via different task prompts. The following works explore the effect of task prompts. Recently, SegGPT [39] extends Painter by learning a generalized one-shot segmentation. In contrast, our method explores the effect of 3D prompts on in-context learning in the point cloud and proposes new baselines for benchmarking 3D in-context learning.

## 3 Method

In this section, we first introduce the task settings of in-context learning in the 3D point cloud, which is motivated by 2D visual in-context learning. Then, we elaborate on the dataset construction and task definitions. Next, based on the MPM framework, we point out the information leakage issue and propose a joint sampling strategy. Finally, we build two distinct baselines using the MPM training methodology.

### 3.1 Modeling In-Context Learning in 3D Point Cloud

**In-context Learning in 2D.** During training, the 2D in-context learning models [4] take two pairs as inputs, including reference pair (or task prompt pair), $R_i^k = \{I_i, T_i^k\}$ and query inputs, $Q_j^k = \{I_j, T_j^k\}$, where $R_i^k$ is the task prompt containing one image $I_i$ and one target $T_i^k$. $Q_j^k$ represent current input image $I_j$. Here, $k$ represents the task index while $i$ and $j$ indicate different example indexes. When $i = j$, we term the prompt as an ideal prompt. During training, both Visual Prompt [4] and Painter [38] combine two pairs of images that perform the same task into a grid-like image and randomly mask portions, following MAE [13]. During the inference stage, only example pairs and a query image are provided. They are combined into a grid-like image with a quarter mask to mask the $T_j^k$, and a pre-trained model is used to restore the missing parts.

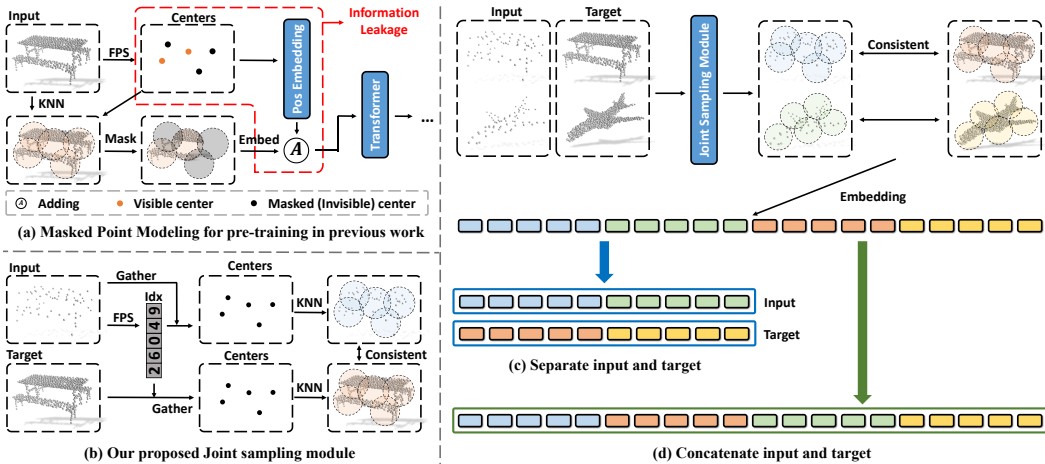

(a) Masked Point Modeling for pre-training in previous work

(b) Our proposed Joint sampling module

(c) Separate input and target

(d) Concatenate input and target

Figure 2: **(a) The pre-training pipeline used in previous works.** When performing Masked Point Modeling (MPM), these works [44, 26, 45] use the center position of the target patches for position embedding, which results in information leakage. **(b) Joint Sampling module.** When selecting the center points for the input and target point clouds, the same indexes are used, which are sampled from the input point cloud. **(c)(d) Different combination forms of input and target point clouds,** which respectively denote the input form of our two baselines, PIC-Sep and PIC-Cat.

**In-context Learning in 3D Point Cloud.** Motivated by 2D in-context learning, we design a similar procedure for 3D in-context learning. During training, each input sample contains two pairs of point clouds that perform the same task as in 2D-context learning. Each pair consists of an input point cloud and its corresponding output point cloud for the given task. Similar to PointMAE [26], we adopt the farthest point sampling and K-nearest neighbor (KNN) techniques to convert the point clouds into a sentence-like data format. These point patches are subsequently encoded into tokens. During the inference, the input point cloud is a combination of example input and query point cloud, while the target point cloud consists of an example target along with masked tokens, as shown in Fig. 1(c). Based on different $R_i^k$, given input $P_j^k$, the model outputs a corresponding target $T_j^k$.

## 3.2 Dataset and Tasks Definition

**ShapeNet In-Context Datasets.** Since there is no previous benchmark for 3D in-context learning, in order to establish the first benchmark for 3D in-context learning, we carefully curate datasets and define task specifications. Firstly, we obtain samples from publicly available datasets, such as ShapeNet [7], ShapeNetPart [42] and transform them into the "input-target" format as stated in Sec.3.1. Additionally, to augment the sample size for the part segmentation task, we conduct several random operations, including point cloud perturbation, rotation, and scaling, on the ShapeNetPart. Consequently, we construct an extensive dataset encompassing all four types of tasks (mentioned below), comprising 217,454 samples. Each sample comprises an input point cloud and its corresponding target for a specific task. Then, we standardize the inputs and outputs to solely contain only the point cloud's XYZ coordinates. For reconstruction and denoising tasks, our aim is to create a clean and aligned point cloud; whereas for the registration task, our aim is to create a clean, registered point cloud. Additionally, the output of the part segmentation task is several point clusters, representing different parts of the object.

**Reconstruction.** The objective of this task is to reconstruct a complete dense point cloud using only a sparse set of points. To evaluate the model's reconstruction capability, we establish five levels for input point clouds, which contain 512, 256, 128, 64, and 32 points respectively.

**Denoising.** In this task, the input consists of a point cloud with Gaussian noise. The objective is to remove the noise surrounding the point cloud, resulting in a clear and distinct object shape. To evaluate the model's performance across different noise levels, we establish five noise levels ranging from 100 to 500 noisy points.

**Registration.** The objective of this task is to restore a rotated point cloud to its original orientation. It is assumed that during both training and inference, the query point cloud and prompt are synchronized in terms of the rotation angle. To avoid coupling with the outputs of the denoising and reconstruction

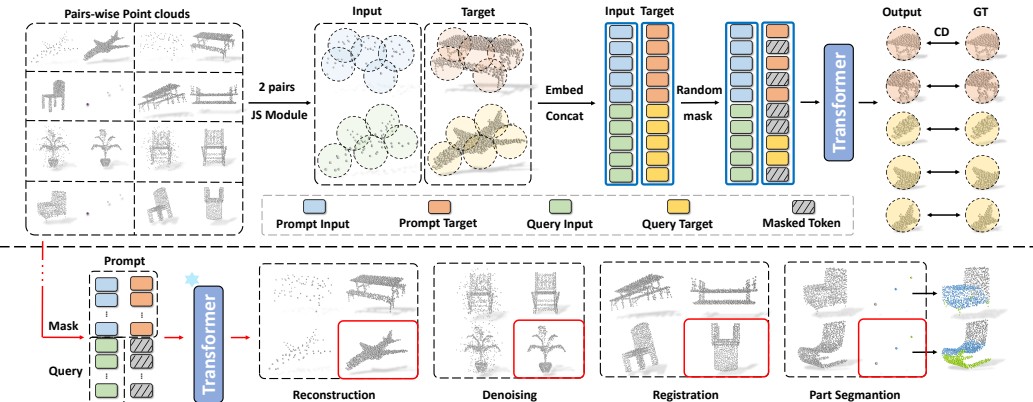

Figure 3: **Overall scheme of our Point-In-Context**. *Top*: Training pipeline of the Masked Point Modeling (MPM) framework. During training, each sample comprises two pairs of input and target point clouds that tackle the same task. These pairs are fed into the transformer model to perform the masked point reconstruction task, which follows a random masking process. ***Bottom***: In-context inference on multitask. Our Point-In-Context could infer results on various downstream point cloud tasks, including reconstruction, denoising, registration, and part segmentation.

tasks, we set the registration task's output as both an upright and an upside-down point cloud.To provide a comprehensive evaluation, we define five levels for the registration task. As the level increases, the range of optional rotation angles also increases.

**Part Segmentation.** The goal of this task is to segment an object into several components, typically 2-6 components. Conventionally, a $C$-dimensional one-hot code is assigned to every point to classify them, where $P$ is the total count of categories. However, for in-context learning, we need to keep the input and output in the same space, which only contains XYZ coordinates, making it a regression task. To achieve this, we convert the $P$ component labels to $P$ discrete points containing XYZ coordinates that are uniformly distributed within a cube. Thus, the output of this task is clusters of points.

### 3.3 Point-In-Context Models

**MPM For 3D In-Context Learning.** Following previous works [4, 38], we adopt the masked point modeling framework for point clouds and propose Point-In-Context (PIC), where we treat the points in MPM as image tokens in MIM, allowing us to leverage the transformer for processing both types of data. As shown in Fig 3, during training, we aim to reconstruct the masked points, aided by the input and visible target 3D points. During the inference stage, given query point clouds as input, depending on the task prompted by the example pair, it can generate the corresponding target, such as reconstruction, denoising, registration, or part segmentation outputs of the query point cloud.

**Information Leakage.** Though MPM [26, 45] exists as a base framework for us, simply adapting it to point clouds is not feasible. As shown in Fig. 2(a), previous pre-training pipelines embed positional information with the center point coordinates of all patches, even for those that are masked out (invisible). Since the patches masked out in the target are invisible in our setting, such an operation will cause information leakage, which does not satisfy the requirements. Furthermore, we find that the sine-cosine encoding sequence will significantly reduce the model performance compared to learned embedding, and even lead to the collapse of the training. The reason is that valuable position information is missing, making it impossible for the model to locate the patches that need to be reconstructed during the processing. Unlike 2D images, 3D point cloud patch sequences have no fixed position (unordered property [27]), so we need to align the patch sequences generated from the input and target point clouds.

**Joint Sampling (JS) Module.** To handle the above issues, we collect $N$ central points from each input point cloud and retrieve their indexes, which we then use to obtain the central points of every patch in both the input and target point clouds. The process is shown in Fig. 2(b). The key of our JS module is the consistency between center point indices of corresponding patches in both target and input point clouds. In other words, the order of the input token sequence and the target token sequence are well-aligned. Such a design compensates for the missing positional embedding of the

Table 1: Comparison of task-specific, multitask, and in-context learning models on four 3D point cloud tasks. For reconstruction, denoising, and registration, we report Chamfer Distance [11] loss (x1000). For part segmentation, we report mIOU.

| Models | Venues | Reconstruction CD ↓ | | | | | | Denoising CD ↓ | | | | | | Registration CD ↓ | | | | | | Part Seg. |
|---|---|---|---|---|---|---|---|---|---|---|---|---|---|---|---|---|---|---|---|---|
| | | L1 | L2 | L3 | L4 | L5 | Avg. | L1 | L2 | L3 | L4 | L5 | Avg. | L1 | L2 | L3 | L4 | L5 | Avg. | mIOU↑ |
| Task-specific models (trained separately) | | | | | | | | | | | | | | | | | | | | |
| PointNet [27] | CVPR'17 | 3.7 | 3.7 | 3.8 | 3.9 | 4.1 | 3.9 | 4.1 | 4.0 | 4.1 | 4.0 | 4.2 | 4.1 | 5.3 | 5.9 | 6.9 | 7.7 | 8.5 | 6.9 | 77.45 |
| DGCNN [40] | TOG'19 | 3.9 | 3.9 | 4.0 | 4.1 | 4.3 | 4.0 | 4.7 | 4.5 | 4.6 | 4.5 | 4.7 | 4.6 | 6.2 | 6.7 | 7.3 | 7.4 | 7.7 | 7.1 | 76.12 |
| PCT [12] | CVM'21 | 2.4 | 2.4 | 2.5 | 2.6 | 3.0 | 2.6 | 2.3 | 2.2 | 2.2 | 2.2 | 2.3 | 2.2 | 5.3 | 5.7 | 6.3 | 6.9 | 7.2 | 6.3 | 79.46 |
| ACT [9] | ICLR'21 | 2.4 | 2.5 | 2.3 | 2.5 | 2.8 | 2.5 | 2.2 | 2.3 | 2.2 | 2.3 | 2.5 | 2.3 | 5.1 | 5.6 | 5.9 | 6.0 | 7.0 | 5.9 | 81.24 |
| Multitask models: share backbone + multi-task heads | | | | | | | | | | | | | | | | | | | | |
| PointNet [27] | CVPR'17 | 87.2 | 86.6 | 87.3 | 90.8 | 92.2 | 88.8 | 17.8 | 22.0 | 25.6 | 30.4 | 33.2 | 25.8 | 25.4 | 22.6 | 24.9 | 25.7 | 26.9 | 25.1 | 15.33 |
| DGCNN [40] | TOG'19 | 38.8 | 36.6 | 37.5 | 37.9 | 42.9 | 37.7 | 6.5 | 6.3 | 6.5 | 6.4 | 7.1 | 6.5 | 12.5 | 14.9 | 17.9 | 19.7 | 20.7 | 17.1 | 16.95 |
| PCT [12] | CVM'21 | 34.7 | 44.1 | 49.9 | 50.0 | 52.3 | 46.2 | 11.2 | 10.3 | 10.7 | 10.2 | 10.5 | 10.6 | 24.4 | 26.0 | 29.6 | 32.8 | 34.7 | 29.5 | 16.71 |
| Point-MAE [26] | ECCV'22 | 5.5 | 5.5 | 6.1 | 6.4 | 6.4 | 6.0 | 5.6 | 5.4 | 5.6 | 5.5 | 5.8 | 5.6 | 11.4 | 12.8 | 14.8 | 16.0 | 16.9 | 14.5 | 5.42 |
| ACT [9] | ICLR'23 | 7.4 | 6.6 | 6.5 | 6.6 | 7.0 | 6.8 | 7.3 | 6.8 | 7.0 | 6.8 | 7.2 | 7.0 | 12.2 | 14.4 | 19.4 | 25.5 | 29.0 | 20.1 | 12.08 |
| I2P-MAE [47] | CVPR'23 | 17.0 | 16.0 | 16.7 | 17.2 | 18.5 | 17.2 | 20.6 | 20.4 | 20.1 | 18.3 | 18.8 | 19.6 | 32.5 | 31.3 | 31.1 | 31.6 | 31.2 | 31.5 | 22.60 |
| ReCon [29] | ICML'23 | 12.4 | 12.1 | 12.4 | 12.5 | 13.1 | 12.5 | 20.4 | 24.5 | 27.2 | 29.2 | 32.5 | 26.9 | 14.7 | 16.3 | 19.2 | 21.5 | 22.5 | 18.8 | 7.71 |
| In-context learning models | | | | | | | | | | | | | | | | | | | | |
| Copy | | 155 | 153 | 152 | 156 | 155 | 154 | 149 | 155 | 157 | 155 | 155 | 154 | 155 | 157 | 156 | 148 | 154 | 154 | 24.18 |
| Point-BERT [44] | CVPR'22 | 288 | 285 | 292 | 286 | 308 | 292 | 292 | 293 | 298 | 296 | 299 | 296 | 291 | 295 | 294 | 295 | 298 | 294 | 0.65 |
| Our PIC-Cat | | **3.2** | **3.6** | 4.6 | 4.9 | **5.5** | **4.3** | **3.9** | **4.6** | **5.3** | **6.0** | **6.8** | **5.3** | 10.0 | 11.4 | 13.8 | 16.9 | 18.6 | 14.1 | **78.95** |
| Our PIC-Sep | | 4.7 | 4.3 | **4.3** | **4.4** | 5.7 | 4.7 | 6.3 | 7.2 | 7.9 | 8.2 | 8.6 | 7.6 | **8.6** | **9.2** | **10.2** | **11.3** | **12.4** | **10.3** | 74.95 |

target while avoiding information leakage. Therefore, it facilitates the model to learn the inherent association between input and target and streamlines the learning process. Subsequently, all point clouds search for neighborhoods containing $M$ points based on the center points corresponding to each patch.

**Point-In-Context Model Architecture.** We use a standard transformer with an encoder-decoder structure as the backbone of our Point-In-Context, and a simple $1 \times 1$ convolutional layer as the task head for point cloud reconstruction. Inspired by Painter [38] and MAE [13], we explore two different baselines for PIC and name them PIC-Sep and PIC-Cat. For PIC-Sep, we take the input and masked target point clouds parallel to the transformer and then merge their features after several blocks, using a simple average for the fusion operation. For PIC-Cat, we concatenate the input and target to form a new point cloud. Then we mask it globally and feed it to the transformer for prediction. We denote the prompt pair as $R_i^k = \{P_i, T_i^k\}$ and query inputs, $Q_j^k = \{P_j, T_j^k\}$, then PIC-Sep and PIC-Cat can be formalized as:

$$P^{\text{Sep}} = \text{Transformer}([P_i \parallel P_j], ([T_i^k \parallel T_j^k], M)), \tag{1}$$

$$P^{\text{Cat}} = \text{Transformer}([I_i \parallel T_i^k \parallel I_j \parallel T_j^k], M), \tag{2}$$

where $\parallel$ is the concatenate operation, and $M$ is the masked token to replace the invisible token. These two input forms are shown in Fig. 2(c)(d).

**Loss Function.** The model is trained to reconstruct the masked point patches. To this end, we use the $\ell_2$ Chamfer Distance as the training loss. Specifically, we calculate the Chamfer Distance between each predicted patch $P$ and its corresponding ground truth $G$.

$$\mathcal{L}(P, G) = \sum_{p \in P} \min_{g \in G} \|p - g\|_2^2 + \sum_{g \in G} \min_{p \in P} \|p - g\|_2^2 \tag{3}$$

## 4   Experiments

**Implementation Details.** We sample 1024 points of each point cloud and divide it into $N = 64$ point patches, each with $M = 32$ neighborhood points. We set the mask ratio as 0.7. For PIC-Sep, we merge the feature of input and target at the third block. We randomly select a prompt pair that performs the same task with the query point cloud from the training set. We use an AdamW optimizer [23] and cosine learning rate decay, with the initial learning rate as 0.001 and a weight decay of 0.05. All models are trained for 300 epochs.

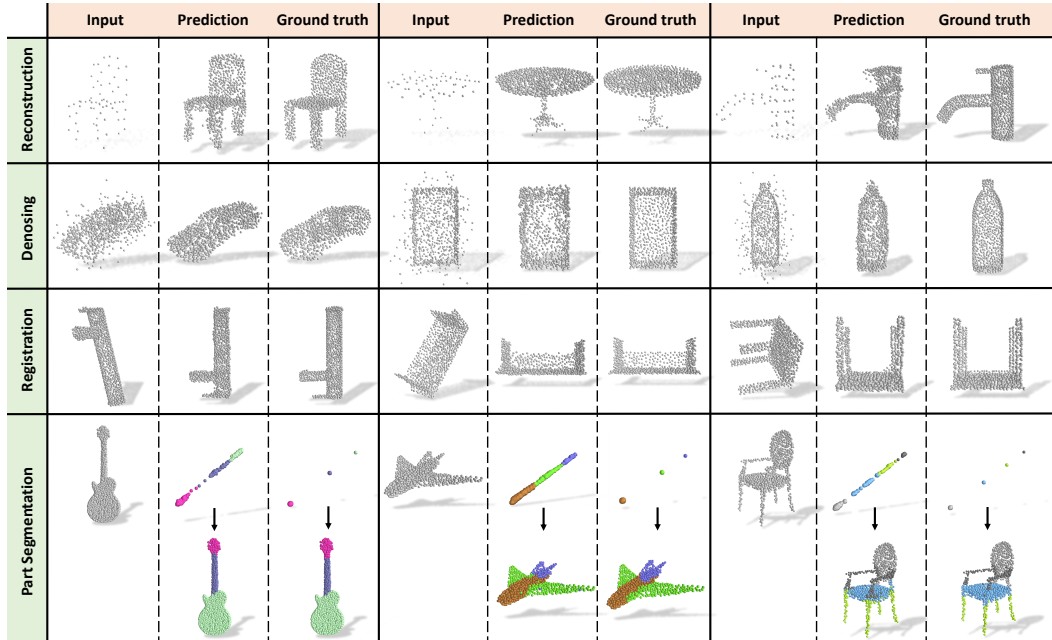

Figure 4: **Visualization of predictions obtained by our Point-In-Context and their corresponding targets in different tasks**, such as reconstruction, denoising, registration, and part segmentation. For part segmentation, we visualize the generated target together with the mapping back, both adding category-specific colors for better comparisons.

## 4.1 Baseline Methods and Training Details

To evaluate the performance of the proposed framework, we evaluate the two variants PIC-Sep and PIC-Cat on the dataset mentioned in Sec. 3.2, both equipped with the proposed Joint Sampling module. We compare them with the following related methods:

**Point-BERT [44]** is a masked auto-encoder. Like the settings in PIC-Cat, we concatenate 2 pairs of input-target point cloud tokens as a token sequence and input it to Point-BERT, which takes a token sequence as the input and converts them into discrete point tokens from a pre-trained dVAE [40] vocabulary of size 8192.

**Task-specific Models.** We selected three representative methods: PointNet [27], DGCNN [40], PCT [12], and ACT [9], and individually train them on four different tasks mentioned in Sec. 3.2. Additionally, we also designed task-specific heads for each task.

**Multitask Models.** For a fair comparison, we develop a multitask model based on PointNet [27], DGCNN [40], and PCT [12], respectively, which are capable of multitask learning. These models feature a shared backbone network and task-specific heads designed to address the needs of different tasks. This design allows simultaneous learning of all four tasks.

**Point-MAE [26]** is a masked auto-encoder. Unlike Point-BERT, Point-MAE directly rebuilds points in each local area. **ACT [9]**, **I2P-MAE [47]**, and **ReCon [29]** are recent SOTA methods that involve other modalities like image and text knowledge in the pre-training stage and enhance the performance on different tasks after fine-tuning the models. Similar to multitask models, we use a pre-trained encoder and combine it with different task heads for simultaneous training on the four tasks.

**Copy Example** is a baseline that utilizes the target point cloud of the prompt as its prediction.

## 4.2 Main Results

We report extensive experimental results of various models on the dataset we proposed in Tab. 1. From where, we found that our PIC-Cat and PIC-Sep exhibit impressive results and are capable of adapting to different tasks after only one training, achieving state-of-the-art results in all four tasks amount multitask models. Besides, we visualize the in-context 3D inference results of PIC-Sep in

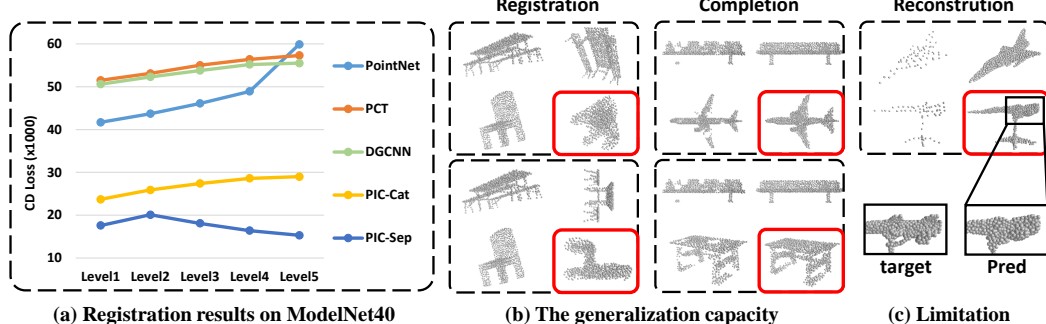

| Registration | Completion | Reconstruction |

(a) Registration results on ModelNet40 | (b) The generalization capacity | (c) Limitation

Figure 5: **(a) Generalized to out-of-distribution data.** We evaluate the registration task on the ModelNet40 [41] dataset, which is not present in the training set. The Chamfer Distance (x1000) is used as the evaluation metric. **(b) Generalized to new tasks.** Our model is able to rotate at any angle according to the example pairs and show the completion ability to the broken point cloud, which is not present in the training set. **(c) Limitation of our model.** PIC cannot reconstruct the detailed parts of complex point clouds very well.

Fig. 4, where our model can generate the corresponding predictions given the provided prompts among all four tasks, including reconstruction, denoising, registration, and part segmentation.

**Comparison to Task-specific Models.** Task-specific models including PointNet [27], DGCNN [40], and PCT [12] outperform PIC-Cat and PIC-Sep in most indicators on all tasks. Specifically, in the part segmentation task, PIC-Cat achieves better results than PointNet and DGCNN, though they are well-designed for segmentation. It needs to be clarified that the performance of our model is directly related to the choice of prompts. When the quality of the prompt is better, PIC-Sep and PIC-Cat can achieve much better results, and this will be demonstrated in the following section.

**Generalization.** Usually, in-context Learning has a certain degree of generalization ability, thus allowing the model to quickly adapt to different tasks. This also applies to our models. First, we test the proposed methods on out-of-distribution point clouds by conducting a registration evaluation on the ModelNet40 [41] dataset. As shown in Fig. 5 (a), our models, PIC-Sep and PIC-Cat, both exhibit superior performance on open-class tasks compared to supervised learning models specifically trained for the registration task. We compare it with models trained on the single task, such as PointNet [27], DGCNN [40], and PCT [12]. These models suffer obvious performance drops when transferred to the new dataset. The more difficulty (higher level of rotating), the more drops.

Furthermore, we validate their generalization abilities on unseen tasks, such as registration and local completion. As shown in Fig. 5 (b), our proposed models work well on both tasks, validating their abilities to transfer learned knowledge. As a comparison, task-specific models are unable to infer the rotated point cloud unless trained with clear supervision. Besides, recovering a local hole is also an impossible task for them, even if they are trained in the reconstruction task.

### 4.3 Analysis

**Effectiveness of Joint Sampling.** We investigate whether the JS module has a valid effect on four tasks. As shown in Tab. 2, both PIC-Sep and PIC-Cat face a sharp decline in performance on the four tasks when the JS module is absent, and they even fail to achieve the most basic goal of reconstructing masked tokens. These findings validate our intuition that maintaining the consistency of the input and target token sequence positions is an indispensable design. That is to say, the JS module is a simple yet effective way to compensate for the missing positional information.

Table 2: Effectiveness of JS Module.

| Model | JS | Den. CD↓ | Part Seg. mIOU↑ |
|---|---|---|---|
| PIC-Cat | ✗ | 29.3 | 17.03 |
| | ✓ | **5.3** | **78.95** |
| PIC-Sep | ✗ | 36.3 | 23.72 |
| | ✓ | **7.6** | **74.95** |

**Ablation on Point Sampling.** We study how the sampling method used by PIC-Sep in JS modules affects the performance of the model. We use the two most common sampling methods: Farthest Point Sampling (FPS) [27], and Random Sampling (RS) [14]. As shown in Tab. 3 (a), for the tasks of reconstruction, denoising, and registration, FPS produced better results than RS. Especially for

Table 3: Ablation study on our Point-In-Context. Gray: default setting

(a) Different sampling strategies.

| # | Sample method | Rec. CD↓ | Den. CD↓ | Reg. CD↓ | Part Seg. mIOU↑ |
|---|---|---|---|---|---|
| 1 | RS | 42.6 | 11.6 | 12.1 | **77.24** |
| 2 | FPS | **4.7** | **7.6** | **10.3** | 74.95 |

(b) Different loss functions.

| # | Loss function | Rec. CD↓ | Den. CD↓ | Reg. CD↓ | Part Seg. mIOU↑ |
|---|---|---|---|---|---|
| 1 | $\ell_1$ | 5.0 | 8.1 | 11.1 | 72.35 |
| 2 | $\ell_2$ | **4.7** | **7.6** | **10.3** | **74.95** |
| 3 | $\ell_1 + \ell_2$ | 5.3 | 7.9 | 13.3 | 70.46 |

(c) Prompt position.

| # | order | Rec. CD↓ | Den. CD↓ | Reg. CD↓ | Part Seg. mIOU↑ |
|---|---|---|---|---|---|
| 1 | behind | 4.8 | 8.2 | **8.0** | 74.04 |
| 2 | before | **4.7** | **7.6** | 10.3 | **74.95** |

point cloud reconstruction, where FPS can collect more key points than RS, and these key points can describe the original outline of the entire point cloud. However, in the task of part segmentation, the results of RS exceed those of FPS.

**Loss Function.** We conduct an exploration to determine which loss function is most suitable for our Point-In-Context model. During training, we experiment with using $\ell_1$, $\ell_2$, and a combination of $\ell_1$ and $\ell_2$ as the loss functions for our PIC-Sep. As Tab. 3 (b) shows, $\ell_2$ achieves the best result on all four tasks mentioned above.

**Prompt Engineering.** We investigate the influence of the layout of the prompt and the query on the experimental results. For PIC-Sep, we set up two layout options: one with the prompt before the query, and the other with the prompt after the query. As shown in Tab. 3 (c), the performance between the two designs has neglectable differences. We simply choose the "before" option to align with 2D in-context learning works.

**Mask Ratio.** We conduct ablation experiments on the mask ratio at a wide range (20%-70%). As shown in Tab 4, training our Point-In-Context with a lower mask ratio weakens its performance across various tasks, especially on the mask ratio 20%. Meanwhile, the best results in the four tasks are distributed at different mask ratios, but considering all downstream tasks as a whole, the model can achieve the highest performance when the mask ratio is 70%. Different from language data, we also find keeping sparsity in training is necessary for mask point modeling for in-context learning. We find similar results as in MAE [13], a higher mask ratio is required to make sure that the model can learn hidden features well.

Table 4: Ablation study on mask ratio.

| # | Mask Ratio | Rec. CD↓ | Den. CD↓ | Reg. CD↓ | Part Seg. mIOU↑ |
|---|---|---|---|---|---|
| 1 | 0.2 | 27.8 | 33.5 | 68.2 | 43.84 |
| 2 | 0.3 | 5.2 | **7.3** | 14.8 | 56.72 |
| 3 | 0.4 | 5.0 | 7.4 | 12.3 | 60.25 |
| 4 | 0.5 | **4.0** | 7.5 | 11.5 | 64.68 |
| 5 | 0.6 | 4.9 | 7.8 | **9.4** | 70.17 |
| 6 | 0.7 | 4.7 | 7.6 | 10.3 | **74.95** |

**Prompt Selection.** We explore the impact of the prompt selection on the model's prediction results. For the random selection method, during testing, we randomly select a pair of input-target point clouds from the training set that performs the same task as the query point cloud, serving as a task prompt. For the class-aware selection method, based on the previous one, we further select point clouds belonging to the same category as the query, such as airplanes, tables, chairs, etc. Additionally, we further delve into selecting two alternative prompts. To select pairs of examples that are paired with the query point cloud, we consider two factors: the Chamfer Distance (CD) [11] between the prompt and the query point cloud and the feature similarity between them (features are extracted from pre-trained PointNet [27]), which are respectively denoted

Table 5: Prompt selection methods.

| Model | Selection method | Rec. CD↓ | Den. CD↓ | Reg. CD↓ |
|---|---|---|---|---|
| PIC-Cat | Random | 4.3 | 5.3 | 14.1 |
| | Class-aware | 4.3 | 5.3 | 10.5 |
| | Fea-aware | 4.3 | 5.3 | 10.7 |
| | CD-aware | **4.3** | **5.3** | **9.6** |
| PIC-Sep | Random | 4.7 | 7.6 | 10.3 |
| | Class-aware | 4.6 | 7.4 | 5.1 |
| | Fea-aware | 4.9 | 7.6 | 5.8 |
| | CD-aware | **4.4** | **7.1** | **4.1** |

as CD-aware and Fea-aware. As depicted in Fig. 5, the CD-aware method demonstrates the best performance and even outperforms the task-specific models on the registration task. Note that we report the random method in the main results (Fig. 1), which means our model has a higher ceiling. This provides us a great opportunity to improve downstream task results by selecting higher-quality prompts, which will be the direction of future work.

Figure 6: **Visualization of comparison results between PIC and multitask models.**

**Comparison Results between PIC and Multitask Models.** We conduct a comparison of visualization results between our Point-In-Context models and multitask models on three tasks, including reconstruction, denoising, and registration. It is important to note that the multitask models in this comparison do not utilize the pre-trained backbone. As shown in Fig. 6, compared with other multitask models, our PIC-Sep and PIC-Cat output results are more satisfactory.

**Limitation.** Our experimental results show that our model can adapt to multiple downstream tasks after a single training with the assistance of prompt support. It performs well on the four proposed tasks, indicating its outstanding generalization ability. Nonetheless, our study has inherent limitations. Our model performs conditional generation for all tasks and presents unchallenged performance for concise point clouds. But for point clouds with complex contours, our model struggles and cannot reconstruct the detailed parts of complex point clouds very well, As shown in Fig. 5 (c).

**Board Impact.** Our work is the first to explore in-context learning for 3D point cloud understanding and is a very relevant but underexplored problem, including task definition, benchmark, and baseline models. Besides, we hope the setup of in-context learning in 3D and the curation of the in-context learning dataset is helpful to the community.

## 5 Conclusion

We propose Point-In-Context (PIC), the first framework adopting the in-context learning paradigm for 3D point cloud understanding. Specifically, we set up an extensive dataset of point cloud pairs with four fundamental tasks to achieve in-context ability. We propose effective designs that facilitate the training and solve the inherited information leakage problem. PIC shows its excellent learning capacity, achieves comparable results with single-task models, and outperforms multitask models on all four tasks. Besides, it shows good generalization ability to out-of-distribution samples and unseen tasks and has great potential via selecting higher-quality prompts. We hope it paves the way for further exploration of in-context learning in the 3D modalities.

**Acknowledgements:** This work was supported by the National Natural Science Foundation of China (No. 62203476), and the Natural Science Foundation of Shenzhen (No. JCYJ20230807120801002). This study is also supported under the RIE2020 Industry Alignment Fund Industry Collaboration Projects (IAF-ICP) Funding Initiative, as well as cash and in-kind contributions from the industry partner(s). It is also supported by Singapore MOE AcRF Tier 2 (MOE-T2EP20120-0001). It is also supported by the interdisciplinary doctoral grants (iDoc 2021-360) from the Personalized Health and Related Technologies (PHRT) of the ETH domain.

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

## Supplementary Material

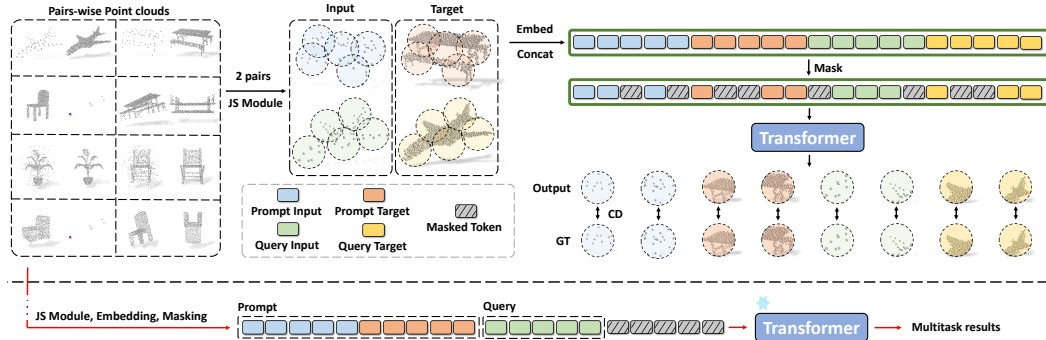

Figure 7: **Overall scheme of our Point-In-Context-Cat**. *Top*: During training, each sample comprises two pairs of input and target point clouds that tackle the same task. Unlike PIC-Sep, PIC-Cat concatenates the input and target to form a new point cloud. ***Bottom***: In-context inference on multitask. Our Point-In-Context could infer results on various downstream point cloud tasks.

**Overview.** The supplementary material includes sections as follows:

- Section A: Pipeline of Point-In-Context-Cat, including training and inference stages.
- Section B: More results about multitask models trained using a pre-trained backbone with multitask heads.
- Section C: More visualization results.

## A More Details of PIC

**Pipeline of PIC-Cat.** During the training phase, our approach involves selecting a pair of query point clouds and a pair of prompt point clouds from the training dataset. These point clouds are then grouped using the Joint Sampling module. Following this, we perform encoding and tokenization on each point cloud and concatenate them to create a new point cloud. A masking operation is applied to the entire point cloud to conduct the MPM task. We set the mask ratio as 60% for our specific approach, PIC-Cat. During the in-context inference stage, we only mask the last quarter of the tokens, which corresponds to the desired output. This approach allows our PIC-Cat model to reconstruct the masked tokens, leveraging its training experience. It is important to note that the task on the query point cloud is determined by the prompt in the example pair.

**Comparison of Model Parameters, GFLOPs, and Test Speed.** We compare the parameters and GFLOPs of each model in the main results of the main text in Tab. 6. Our model achieves a favorable balance between structural complexity and task performance, making it a compelling choice. Note that the parameters and GFLOPs of task-specific models are computed, including four individual models for four different tasks. Besides, we report the speed of models by samples/second tested on one NVIDIA RTX 3080 Ti GPU. Our PIC-Cat presents a high inference speed (953 samples/second), which is second only to DGCNN [40].

Table 6: The comparison of parameters, GFLOPs, and test speed.

|  | Task-specific models | | | multi-task models | | | | In-context learning models | | |
|---|---|---|---|---|---|---|---|---|---|---|
|  | PointNet [27] | DGCNN [40] | PCT [12] | PointNet [27] | DGCNN [40] | PCT [12] | Point-MAE [26] | Point-BERT [44] | PIC-Cat | PIC-Sep |
| Params(M) | 8.9 | 7.9 | 13.0 | 6.0 | 7.6 | 10.8 | 27.0 | 52.6 | 29.0 | 28.9 |
| FLOPs(G) | 1.9 | 3.1 | 6.3 | 1.9 | 10.2 | 6.4 | 11.8 | 12.0 | 12.1 | 8.4 |
| Test speed | 694 | 1500 | 694 | 844 | 1185 | 717 | 742 | 190 | 953 | 291 |

Table 7: Results of multitask models composed of a multitask head and a pre-train backbone trained on ShapeNet [7] for classification. For reconstruction, denoising, and registration, we report Chamfer Distance $\ell_2$ loss (x1000). For part segmentation, we report mIOU.

| Models | Acc.(%) | Reconstruction CD ↓ | | | | | | Denoising CD ↓ | | | | | | Registration CD ↓ | | | | | | Part Seg. |
| | | L1 | L2 | L3 | L4 | L5 | Avg. | L1 | L2 | L3 | L4 | L5 | Avg. | L1 | L2 | L3 | L4 | L5 | Avg. | mIOU↑ |
|---|---|---|---|---|---|---|---|---|---|---|---|---|---|---|---|---|---|---|---|---|
| multitask models: share backbone + multi-task heads | | | | | | | | | | | | | | | | | | | | |
| PointNet [27] | 88.7 | 47.0 | 45.8 | 45.4 | 45.4 | 45.8 | 45.9 | 22.9 | 23.2 | 26.3 | 28.3 | 30.0 | 26.1 | 35.5 | 34.8 | 37.1 | 37.2 | 38.6 | 36.6 | 10.13 |
| DGCNN [40] | 89.4 | 46.7 | 47.2 | 48.1 | 48.6 | 48.5 | 47.8 | 8.2 | 8.3 | 8.4 | 8.8 | 9.2 | 8.6 | 14.2 | 15.8 | 18.2 | 21.8 | 23.5 | 18.7 | 21.35 |
| PCT [12] | 89.5 | 64.7 | 60.8 | 59.2 | 60.1 | 59.7 | 61.0 | 14.5 | 12.2 | 12.4 | 12.0 | 11.8 | 12.6 | 22.6 | 25.2 | 28.3 | 31.1 | 33.2 | 28.1 | 15.43 |

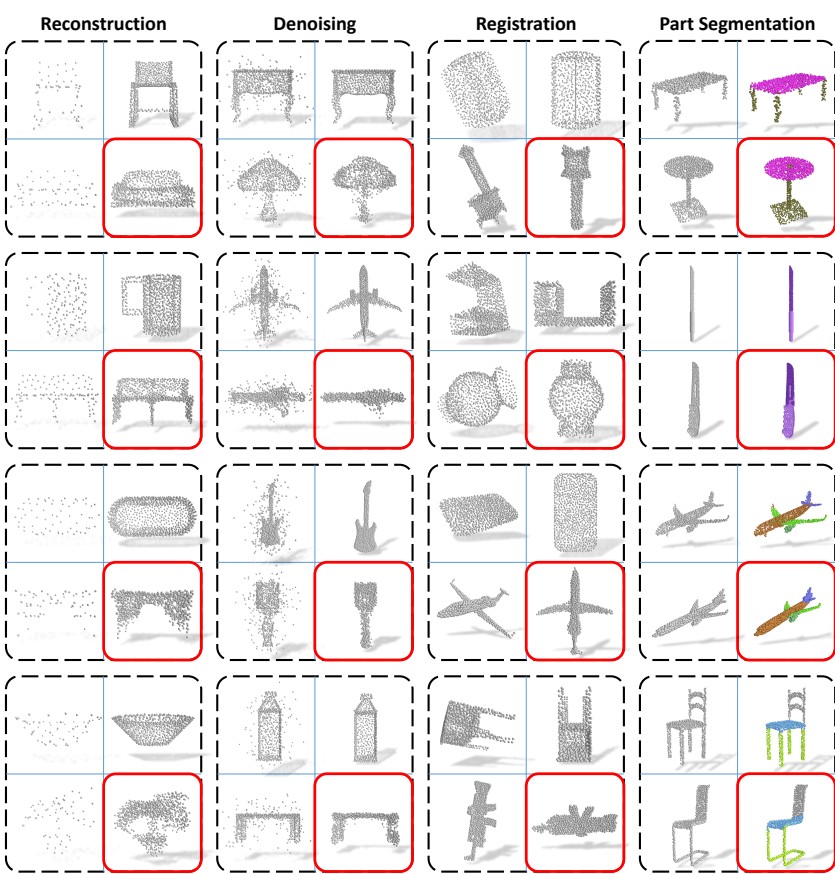

Figure 8: **Additional visualization results of PIC-Sep.** The output of our model is marked in red. Note that the results of part segmentation have been processed by adding XYZ coordinates.

## B  More Results of Multitask Models

**Pre-trained Backbone + Multitask Heads.** For multitask models, we utilize a pre-trained backbone feature extraction network that is trained on the ShapeNet [7] dataset for classification tasks. This pre-trained backbone network is equipped with multiple task-specific heads to perform multitask learning on our benchmark, allowing for the simultaneous handling of various tasks. As shown in Tab. 7, while these supervised models perform well when trained on individual tasks, they exhibit poor performance on multitask benchmarks.

## C  More Visualization

**More Visualization of PIC-Sep.** We visualize more examples in Fig. 8, including reconstruction, denoising, registration, and part segmentation.

