# OpenReview forum: "Explore In-Context Learning for 3D Point Cloud Understanding"
_NeurIPS.cc/2023/Conference — NeurIPS 2023 spotlight_

### Official Review · Reviewer_uxmh · 2023-07-04

**Soundness:** 3 good
**Presentation:** 2 fair
**Contribution:** 2 fair
**Rating:** 5
**Confidence:** 4

**Summary:**

This paper proposes an in-text learning method for multi-task 3D shape analysis. It handles several tasks such as denoising, part segmentation, reconstruction and registration with a single pretrained masked point model (MPM).  The authors also claim previous MPM methods introduce information leakage during pretraining. To solve this, this paper proposes a JS module, which facilitates the model to learn the inherent association between input and target and streamlines the learning process.

**Strengths:**

1.	In-context learning for point clouds is a new and interesting topic.
2.	The unified modeling of different 3D shape analysis tasks makes sense.
3.	The performance of the proposed method is satisfactory.

**Weaknesses:**

1.	The writing of this paper is not clear. (3.1) The authors should add more description on the task definition, such as which kind of model should be pretrained, is it fixed during inference, etc.  (3.3) It is claimed that previous methods will bring information leakage. However, the explanation is confusing. The authors should clearly show this in Figure 2.  Moreover, as a core technical contribution of this paper, JS module should be clearly demonstrated. It is hard for me to understand why using the same FPS Idx.
2.	The JS Module seems to be straight-forward (and confusing). As a simple technique, more insight should be explained.

I will consider to improve my rating if my concern can be properly solved.

**Questions:**

see above

**Limitations:**

see weakness

---

> ### Author Rebuttal · Authors · 2023-08-09
>
> >**Q1**:
> >
> >(1) **Task definition**
> >
> >(2) **Information leakage**
> >
> >(3) **JS module**
>
> **A1**:
>
> &emsp; We are sorry that you were confused about our paper and thank you very much for your suggestions on our paper. We provide more detailed explanations:
>
> &emsp; **1. More details of the task definition in Section 3.1.**
>
> &emsp; **During training**, motivated by 2D in-context learning [Bar et al., 2022; Wang et al., 2022], we provide two pairs of point clouds as input to the model and train the model with an MPM-style framework, which is to reconstruct the masked parts of the target point clouds belonging to two pairs. **During testing**, we provide the prompt pair and the query pair that is comprised of a query point cloud and a mask as input to the model. Then the model will reconstruct the mask like what to do in training, which is the corresponding output to the query point cloud.
>
> &emsp; Besides, Our Point-In-Context is trained from scratch, with **no pre-training nor additional dataset**. As for compared methods, the multi-task models are trained on all tasks jointly, too. But they are comprised of a pre-trained encoder and multi-task heads. The task-specific models are trained on each task individually.
>
> &emsp; Finally, **the model is fixed during inference** with no more learning.
>
> &emsp; **2. The elaboration on the information leakage problem mentioned in Section 3.3.**
>
> &emsp; Previous methods based on the MPM pre-training framework will first be pre-trained on the ShapeNet dataset with Masked Point Modeling. The pre-trained encoder is then combined with task-specific heads to perform fine-tuning on downstream tasks, such as classification, few-shot learning, or segmentation.
>
> &emsp; During pre-training, the model will embed the position of **all sampled local center points** to the extracted point cloud features, even if **they belong to the center points of the patches to be reconstructed (have been masked out, invisible)**. This is possible under their pre-training scheme, but it will cause information leakage under our setting, which **is not allowed**. In other words, our model shouldn't see the masked center points, because we just train one time for multiple tasks and no more fine-tuning.
>
> &emsp; Based on your confusion, we **improve Figure 2(a)** of the main text and put it in **the PDF of the global rebuttal**. Please see the improved figure of the demonstration of information leakage for more details.
>
> &emsp; **3. More explanations and insights for JS modules.**
>
> &emsp; In order to solve the above problems, we replace the position embedding that leads to information leakage with a sin-cos fixed encoding, that is, do not use the local center point for position embedding. However, we found that the performance of the model dropped drastically, and even failed to converge, as shown in the following table.
>
> Caption: The performance (mIOU) of PIC-Sep and PIC-Cat on denoising and part segmentation.
>
> | Model         | Denoising CD$\downarrow$ | Part Seg. mIOU$\uparrow$ |
> |--------------|:---------:|:---------:|
> | PIC-Cat w/o JS |   29.3    |   17.03   |
> | PIC-Cat w/ JS  |   **5.3**     |   **78.95**   |
> |-|-|-|
> | PIC-Sep w/o JS |   36.3    |   23.72   |
> | PIC-Sep w/ JS  |   **7.6**     |   **74.95**   |
>
> &emsp; Unlike images, each point of a point cloud has no clear order structure (**unordered property**), so the position information is seriously missing between the input point cloud and the target point cloud, resulting in the model being unable to learn the input point cloud and the mapping relationship between the target point cloud. Such the unordered property mentioned above is **an unexplored challenge** for introducing in-context learning to 3D point clouds.
>
> &emsp; Therefore, in order to make up for the lack of position information, we **align the indices of input sample pairs for each task during training**. During the **generation of our dataset**, the input point cloud of all tasks is generated from the target point cloud, and different operations are performed for different tasks. Taking the reconstruction task as an example, a sparse input point cloud is obtained by discarding a certain percentage of points in the target point cloud. We zero the XYZ coordinates of the discarded points, so the shape of the input point cloud remains consistent with the target point cloud. **This is why we can make the patch/token sequences of both the input point cloud and the target point cloud well-aligned by using the same FPS index**.
>
> &emsp; For the denoising task, we add Gaussian noise to the points at different levels, so the shape of the input point cloud and the target point cloud are the same and aligned. For the registration task, we multiply the target point cloud with a rotation matrix to get the input point cloud. For the part segmentation task, we convert each point's part label to a point in XYZ space, and points representing the same part are clustered together. All in all, the input and target point clouds are consistent in shape as well as indices of each point for all tasks.
>
> &emsp; Then, we propose a **simple but effective** method, the **Joint Sampling module**, so that each patch/token of the input point cloud and the target point cloud is well-aligned.
>
> &emsp; **During inference**, for the query point cloud, its target is masked out, so **no alignment is required**, we only need to provide the aligned task prompt pair which is randomly sampled in the training set. During our exploration, we tested various positional embeddings, learnable matrices, and sampling methods, but none met our expectations.
>
> [Bar et al., 2022] Visual prompting via image inpainting. In NeurIPS.
>
> [Wang et al., 2022] Images speak in images: A generalist painter for in-context visual learning. In CVPR.
>
> >**Q2: The JS Module seems to be straight-forward (and confusing). As a simple technique, more insight should be explained.**
>
> **A2**:
>
> &emsp; Please see the above explanations.

---

> > ### Author Response · Authors · 2023-08-20
> > **New Comment to Reviewer uxmh**
> >
> > Dear reviewer uxmh,
> >
> > &emsp; Our work is **the first** to explore in-context learning for 3D point cloud understanding. We **unify the output space of the four tasks** into the 3D coordinate space, and apply the MPM framework for model training, realizing **one training, one model, and processing multiple tasks**. Extensive experiment results demonstrate the methodology design's **superiority**, which indicates that in-context learning in 3D is **non-trivial**. Besides, our Point-In-Context shows excellent generalization capability. As the reviewer Uvps says, our work may **bridge the connection between 3D point clouds and 2D images in-context learning**, as we hope it will.
> >
> > &emsp; We would like to have an in-depth discussion with you. Looking forward to your reply.
> >
> > &emsp; Yours sincerely,
> >
> > &emsp; Submission4623 Authors

---

### Official Review · Reviewer_kyHc · 2023-07-06

**Soundness:** 3 good
**Presentation:** 3 good
**Contribution:** 2 fair
**Rating:** 6
**Confidence:** 5

**Summary:**

This paper introduces a novel framework, named Point-In-Context, designed explicitly for in-context learning 3D point clouds. The authors conduct extensive experiments to validate the versatility and adaptability of the proposed methods in handling a wide range of tasks.

**Strengths:**

This paper explores an interesting topic -- how to do in-context learning for 3D understanding; and it demonstrates a reasonable way to achieve it, addresses some issues like the "position information leakage", and shows some positive results.

**Weaknesses:**

The reviewer is concerned about how well this framework can generalize, its general usefulness, and its scope, since to some extent the performance may depend on the "prompt". In reality, it might be hard to choose a proper prompt for it and the performance may not be stable, and it kind of involves some extra tuning effort compared to using the model which is directly trained for that task.

**Questions:**

1. Is the Transformer backbone trained from scratch or it is initialized from something, whether this backbone has been pre-trained on some 3D data before, for example, pointbert, pointmae .etc? Is there any leakage between train dataset and test dataset.
2. Have you tried only train the model for part of the tasks not all of them, and try to see if there is any emergent ability; for example, only train it for reconstruction and registration, do it have some zero-shot emergent ability for denoising and part-segmentation?

**Limitations:**

refer to the weakness part

---

> ### Author Rebuttal · Authors · 2023-08-09
>
> >**Q1: In reality, it might be hard to choose a proper prompt for it and the performance may not be stable, and it kind of involves some extra tuning effort compared to using the model which is directly trained for that task.**
>
> **A1**:
>
> &emsp; The prompt can indeed affect the performance of our PIC, but we consider it to be **the icing on the cake for us**. Moreover, our PIC shows **great generalization ability** which is needed in actual applications.
>
> &emsp; Usually, the prompt plays a very important role in In-context Learning. High-quality prompts can improve the performance of the model. In Table 1 of the main text, the main results of this paper, the prompts we used are randomly selected under the same task in the training set. It can be seen that **the performance brought by these random prompts is already remarkable**, surpassing all multi-task models on the four tasks. However, we have a higher ceiling. When it is possible, **a good prompt shall greatly improve the performance** (see Table 3(a) of the main text and Table 5(a) of the appendix).
>
> &emsp; Besides, we demonstrate that **our PIC has great generalization ability**. In actual scenarios, the 3D domain shows seriously lacking data (as the reviewer GuqQ said), and the data to be tested in actual applications do not necessarily have corresponding training data, so the generalization of the model is more needed. As we mentioned in Figure 5(a) of the main text, when our model generalizes to other datasets (ModelNet40), it can outperform the single-task training model under the same training environment, which shows that **our model can perform well under the limitation of training data in practical applications**. We also demonstrate that our PIC can generalize to other related tasks (see the following A3).
>
> &emsp; In conclusion, we regard **the possibility of choosing a better prompt as an advantage of the proposed method, providing the chance of achieving more reliable performance**. we do not need a “proper” prompt to maintain our performance, the random prompt under the same task is satisfied. Moreover, **our generalization ability can make it more practical in reality compared to task-specific models**.
>
> >**Q2: Is the Transformer backbone trained from scratch or it is initialized from something, whether this backbone has been pre-trained on some 3D data before? Is there any leakage between the training dataset and the testing dataset?**
>
> **A2**:
>
> &emsp; Our model is trained from scratch, **no additional data is used for pre-training**, and **no pre-trained model parameters are used**, all parameters are only updated on our training set. In addition, our training set and test set are divided according to the ShapeNet [Chang et al., 2015] and ShapeNetPart [Yi et al., 2016] datasets **without any leakage**. Our code is reproducible, which is provided in the supplementary material, and the dataset will be made public in the future.
>
> [Chang et al., 2015] Shapenet: An information-rich 3d model repository. In arXiv.
>
> [Yi et al., 2016] A scalable active framework for region annotation in 3d shape collections. In TOG.
>
> >**Q3: Have you tried only training the model for part of the tasks not all of them, and try to see if there is any emergent ability; for example, only train it for reconstruction and registration, does it have some zero-shot emergent ability for denoising and part-segmentation?**
>
> **A3**:
>
> Caption: The performance of PIC-Sep and PIC-Cat on unseen tasks. **Bold represents tasks included in the training set**. For instance, the first row of PIC-Cat denotes that it is trained on reconstruction and registration, and is tested on denoising and part segmentation.
>
>
> | Models        | Reconstruction CD$\downarrow$ | Denoising CD$\downarrow$ | Registration CD$\downarrow$ | Part Seg. mIOU$\uparrow$ |
> |-------------- |:--------------:|:---------:|:------------:|:---------:|
> | PIC-Cat (Original)  |      **4.3**       |    **5.3**    |     **14.1**     |   **78.95**   |
> | PIC-Cat (new)        |     **3.9**       |   19.5    |     **3.9**      |    2.98   |
> | PIC-Cat (new)         |      **4.1**       |   21.0    |     **18.8**     |   **79.51**   |
> |  -  |  -  |  -  |  -  |  -  |
> | PIC-Sep (Original)  |      **4.7**       |    **7.6**    |     **10.3**     |   **74.95**   |
> | PIC-Sep (new)        |      **2.4**       |   25.3    |     **3.9**      |    3.41   |
> | PIC-Sep (new)        |      **6.7**       |   22.9    |     **11.0**     |   **80.46**   |
>
> &emsp; Thanks for your advice. We conduct experiments that train our PIC for part of the tasks and test its emergent ability for the remaining tasks. As the above table shows, we train our PIC for **reconstruction & registration & part segmentation**, and evaluate it on the **denoising** task, which is an unseen task in the training set. Despite the formidable challenge, the PIC demonstrates **great emergent capability on denoising task**.
>
> &emsp; For training on reconstruction & registration and testing on denoising & part segmentation, we don't think it makes sense. Because **the output space of part segmentation is kind of different from the other three, it is hard for PIC to adapt to such a polar task**. Despite it, we conducted the experiment. As we expected, we find that PIC shows excellent generalization capability on the denoising task but performs poor results on the part segmentation task due to the extremely different output space form.
>
> &emsp; It is worth noting that our PIC trained on our dataset **can also generalize to the other dataset (ModelNet40)**, and its performance surpasses the task-specific model trained on our dataset, as shown in **Figure 5(a) of the main text**.

---

> > ### Comment · Reviewer_kyHc · 2023-08-14
> >
> > Thanks for the response, to some extend it addresses my concern, and I will keep my positive score.

---

### Official Review · Reviewer_GuqQ · 2023-07-06

**Soundness:** 2 fair
**Presentation:** 3 good
**Contribution:** 3 good
**Rating:** 6
**Confidence:** 5

**Summary:**

This work is conducted toward in-context learning for 3D point cloud data. Similar to 2D in-context learning, the authors first define and construct the in-context learning 3D dataset covering reconstruction, denoising, registration, and part segmentation tasks. To avoid information leaking during masked point modeling, a joint sampling module that samples correspondingly from inputs and outputs is proposed. Extensive experiments have been conducted, and interesting results have been obtained.

**Strengths:**

- This works targets in-context learning for 3D point clouds, which is a very relevant but underexplored problem.
- The setup of in-context learning in 3D and curation of the in-context learning dataset is helpful to the community.
- Extensive experiments have been conducted, where results demonstrate the superiority of the methodology design, which also indicates that in-context learning in 3D is non-trivial.
- Code and dataset are promised to be released. Good.

**Weaknesses:**

- For 2D or NLP, in-context learning is generally free-form, targeted at various tasks. However, for 3D, due to a seriously lacking of data, it only shows very limited applications. It seems not practical in real-world deployments.
- The novelty and technical contribution of the method part is somewhat limited. The joint sampling module is relatively straightforward. The model and loss function is the same as in previous works. However, the contribution of setting up the 3D in-context learning baseline is good.
- Missing citations or comparison: For 3D MIM methods, important recent cross-modal representation/prompt learning methods should be discussed or compared [Dong et al., 2023; Qi et al., 2023]; For the loss function, Chamfer Distance should be cited [Fan et al., 2017]; For in-context learning, some works are missing [Sun et al., 2023; Balažević et al., 2023].
- The compared methods Point-BERT and Point-MAE are a bit old and are not SOTA. I wonder what about the comparison to other cross-modal 3D MIM methods like ACT [Dong et al., 2023], I2P-MAE [Zhang et al., 2023], and ReCon [Qi et al., 2023]? I think it is important to conduct solid comparisons to more advanced methods.
- Minor suggestion: There are not many formulations but the current formulation is not neat. Please improve the presentation quality including the formulations. For example, pure texts should not be italic in equations. Besides, the writing is overall okay but could be improved.

[Dong et al., 2023] Autoencoders as Cross-Modal Teachers: Can Pretrained 2D Image Transformers Help 3D Representation Learning? In ICLR.

[Qi et al., 2023] Contrast with Reconstruct: Contrastive 3D Representation Learning Guided by Generative Pretraining. In ICML.

[Fan et al., 2017] A point set generation network for 3d object reconstruction from a single image. In CVPR.

[Sun et al., 2023] Exploring Effective Factors for Improving Visual In-Context Learning. arXiv preprint.

[Balažević et al., 2023] Towards In-context Scene Understanding. arXiv preprint.

[Zhang et al., 2023] Learning 3d representations from 2d pre-trained models via image-to-point masked autoencoders. In CVPR.

**Questions:**

Besides questions and concerns listed before, I have following questions:
- Is all tasks jointly trained or one model per task?
- What if the 3D in-context learning involves other modalities? For example, images or languages?
- In ACT [Dong et al., 2023], I notice that the authors report the reconstruction CD-$\ell_2$ on ShapeNet as 2.110, which is significantly lower (better) than the best reconstruction results in this paper (4.3). Can authors explain this?

I am looking forward to the author's response, and I am happy to raise my score if my concerns are solved.

**Limitations:**

The authors have discussed the limitations, and I think it is somewhat okay for this work.

---

> ### Author Rebuttal · Authors · 2023-08-09
>
> Due to word limit, we omit the questions, answer order is consistent with the above questions.
>
> **A1**:
>
> &emsp; Seriously lacking data is a **common problem for 3D tasks**. With the advent of 3D sensors such as LiDAR and Kinect, 3D point clouds have gained increasing popularity and are widely used in robotics, autonomous driving, object detection, etc.
>
> &emsp; First, our work is **the first to explore the application of in-context learning in 3D point clouds and goes one step further than previous methods**.
>
> &emsp; Secondly, our PIC can **easily be generalized to related datasets (Figure 5(a) of the main text) and tasks (A3 of the reviewer kyHc)**. We strive to tap the multi-task potential of the model as much as possible under the condition of fewer data (can see **A1 of the reviewer kyHc**) and provide a new idea and direction for future work of 3D point clouds. Such capability is urgently needed for a deep learning model in the condition of a seriously lacking 3D point cloud data.
>
> &emsp; In conclusion, the proposed method **is good at solving the lacking data problem**.
>
> **A2**:
>
> &emsp; Our work is **the first to explore in-context learning for 3D point cloud understanding**. We unify the output space of the four tasks into the 3D coordinate space, and apply the MPM framework for model training, realizing one training, one model, and processing multiple tasks.
>
> &emsp; It is **a challenge to train multiple tasks jointly**. Previous methods cannot achieve an ideal balance among multiple tasks, resulting in poor multi-task performance, as shown in Table 1 of the main text. Our PIC shows a great ability for dealing with multiple tasks jointly with an MPM-style training scheme.
>
> &emsp; However, it is infeasible to simply adopt the MPM framework into our work due to **information leakage**. That is the model knows the coordinates of the center point of the patch to be reconstructed in advance, which **is not allowed**. It is an open problem that is never explored.
>
> &emsp; Therefore, in order to make up for the lack of position information, we propose a simple but effective method, the **Joint Sampling module**, so that each patch/token of the input point cloud and the target point cloud is well-aligned. **During our exploration**, we tested various positional embeddings, learnable matrices, and sampling methods, but none met our expectations.
>
> &emsp; As for the model structure, we improve some detailed architecture for our different forms of inputs. But what we want to express most is that **we are the first work to introduce in-context learning into the 3D point clouds, laying the foundation for subsequent research**.
>
> **A3**:
>
> &emsp; Thank you for your valuable suggestions. We will cite these articles in the final version and add the comparison of ACT, I2P-MAE, and ReCon to the main experimental results.
>
> **A4**:
>
> &emsp; Thank you for your constructive suggestions, which make our work more complete and solid. We conduct experiments on other cross-modal methods like ACT, I2P-MAE, and ReCon. In the implementation, we use a pre-trained encoder and combine it with different task heads for simultaneous training on the four tasks. As shown in the table of **the PDF of the global rebuttal**, our PICs outperform them on four tasks. Such results demonstrate that our PICs show an excellent ability to deal with multi-task. **We will include these results in the final version**.
>
> **A5**:
>
> &emsp; Thanks for your valuable advice. Except for the typo you have mentioned, we will carefully check all formulas and polish the expression of the article in the final version to improve its readability.
>
> **A6**:
>
> &emsp; Our Point-In-Context is trained on all tasks jointly. As for compared methods, the multi-task models are trained on all tasks jointly, too. The task-specific models are trained on each task individually. The joint training signifies the proposed method that is capable of learning various contexts and bringing mutual benefits.
>
> **A7**:
>
> &emsp; It's an exciting idea, and we consider it feasible to involve the other two modalities. Our work is the first to explore in-context learning for 3D point cloud modality. Similar to the ICL in language and 2D images, our model is based on transformer architecture, providing us a chance to involve the other two modalities.
>
> &emsp; The main challenge is that there are not many data sets with three modalities and data alignment, and how to **align the features** of the three modalities to the same feature space is a semi-open problem. There are already several works exploring the multimodal understanding of 3D scenes (PLA [1] and Open Scene [2], etc.), and future work will focus on exploring the in-context learning in 3D point cloud scene segmentation. Please see **A2 of reviewer Uvps** for more details.
>
> [1] PLA: Language-Driven Open-Vocabulary 3D Scene Understanding. CVPR2023.
>
> [2] OpenScene: 3D Scene Understanding with Open Vocabularies. CVPR2023.
>
> **A8**:
>
> &emsp; The reconstruction result of 2.110 is an ablation experiment in the ACT paper, and we did not find more details about this experiment in the paper. Therefore, we speculate that there are two reasons:
>
> &emsp; **1. The settings of the experiments are different.** We establish five levels for input point clouds, which contain 512, 256, 128, 64, and 32 points respectively for our reconstruction task. The difficulty of our reconstruction task cannot be measured and compared with the reconstruction task in the ACT directly.
>
> &emsp; **2. About the pre-processing of experimental data.** In order to unify the data between different tasks and different datasets, we normalize the data, resulting in the point clouds of the testing set in our reconstruction dataset being **larger** than the original data (see **table R2 in PDF of the global rebuttal**). So our results may be seen as higher than the results of ACT intuitively. However, our results cannot be compared with the result of the ACT paper directly.

---

> > ### Comment · Reviewer_GuqQ · 2023-08-12
> > **Post Rebuttal Comment**
> >
> > I thank the authors for the detailed response! My concerns are largely solved. Thus, I raise the score to Weak Accept.
> >
> > Minor question: I have checked the part segmentation results again, and I found that the results are lower than commonly reported numbers on ShapeNetPart. For example, PointNet has only 77.45 mIoU, which is lower than the common result of 80.39 mIoU. This situation goes to other methods like DGCNN,  ACT, etc. Is the result tested on the same ShapeNetPart? Or is it a newly constructed dataset split?

---

> > > ### Author Response · Authors · 2023-08-13
> > > **The response to the reviewer's question.**
> > >
> > > &emsp; Thank you for your nice comment!
> > >
> > > &emsp; Our testing set of part segmentation is **different from** ShapeNetPart [Li et al., 2016]. Ours is a newly constructed dataset, which is **larger** and **more difficult** than ShapeNetPart.
> > >
> > > &emsp; As described in Section 3.2 of the main text, to augment the sample size of the part segmentation task, we conduct several random operations on the point clouds of the ShapeNetPart dataset, including point cloud perturbation, rotation, and scaling. Therefore, the size of the testing set of our part segmentation task is about 4 times larger and more diverse than that of ShapeNetPart, and it is more difficult compared to ShapeNetPart which contains regular point clouds, therefore our results are lower than commonly reported numbers on ShapeNetPart.
> > >
> > > >[Li et al., 2016] A scalable active framework for region annotation in 3d shape collections. In TOG.

---

> > > > ### Comment · Reviewer_GuqQ · 2023-08-13
> > > > **Confusion Addressed**
> > > >
> > > > Thanks for the response. My confusion has been addressed.

---

### Official Review · Reviewer_Uvps · 2023-07-07

**Soundness:** 3 good
**Presentation:** 4 excellent
**Contribution:** 3 good
**Rating:** 7
**Confidence:** 5

**Summary:**

Inspired by in-context learning in NLP and 2D vision tasks, this paper aims to explore the in-context learning in the 3D point cloud. The authors present Point-In-Context, which is a 3D mask point modeling framework. Meanwhile, to handle the data leakage issues, the authors also present a simple solution, named joint sampling. Then, extensive experiments are carried out in modified ModelNet40 dataset. The performance of two different baselines is good to other specific baselines.

**Strengths:**

1. The proposed point in-context is the first work that explores the in-context ability in the point cloud understanding, which is novel and interesting to me. The authors show the effectiveness on four different tasks, including rotations, registration, denoising and part segmentation.

2. The overall writing and motivation is good, clear, and easy to follow.

3. The proposed joint sampling can effectively solve the data leakage problems.

4. The experiments results are good. The ablation studies are extensive. The authors re-benchmark several representative works, including SOTA single model and multi-task models. The analysis on visual point example is good and convincing.

5. This work may bridge the connection between 3D point cloud and 2D image in-context learning.


**Weaknesses:**

1. Are there any other solutions to replace joint sampling to handle the data leakage problems?

2. The proposed approach is verified effective on simple scene on ModalNet40. The ability to extend to large scale scene including in-door point segmentation is unknown.

3. What are the results of two combined prompts: denoising and part segmentation jointly?

**Questions:**

Please refer to the *Weaknesses.

**Limitations:**

Yes.

---

> ### Author Rebuttal · Authors · 2023-08-09
>
> >**Q1: Are there any other solutions to replace joint sampling to handle the data leakage problems?**
>
> **A1**:
>
> &emsp; When we adopt the original MPM pre-training framework for our task, we find that model utilizes center point coordinates that should have been masked before position embedding. This inadvertent inclusion results in unintended information leakage.
>
> &emsp; This problem is **not explored in previous work and is an open question**. To address this predicament, we introduce the **Joint Sampling module** designed to align the positional attributes of patches/tokens between input and target point clouds. This alignment serves to compensate for the consequential loss of vital positional data, thereby enabling the model to understand the intricate mapping correlation inherent in the input-target pairs.
>
> &emsp; During our exploration, we tested various positional embeddings, learnable matrices, and sampling methods, but none met our expectations. In the end, our **simple yet highly effective** Joitn Sampling module provided the perfect solution to this challenge. In future work, we will consider **enhancing** the JS module's structure by incorporating a learnable matrix. This will strengthen the linkage between the input point clouds and the target point clouds, thereby advancing overall performance and efficacy.
>
> >**Q2: The ability to extend to large-scale scenes including in-door point segmentation is unknown.**
>
> **A2**:
>
> &emsp; Segmenting scene-level point clouds presents a formidable challenge. The 3D point cloud datasets comprise an extensive number of points and contain intricate object composition, such as ScanNet [Dai et al., 2017], Matterport3D [Chang et al., 2017], and S3DIS [Armeni et al., 2016].
>
> &emsp; To solve these problems, more fine-grained design and more computing resources are required. However, we think that **this is feasible**. Referring to PointNet and others' works, we can divide the 3D scene into blocks using a non-overlapping sliding window of 1m x 1m on the xy plane. By processing these blocks individually and subsequently merging their results, we can derive the ultimate segmentation result of a whole 3D scene. Naturally, if an ample supply of computational resources is available, a comprehensive analysis of the entire scene can yield a deeper understanding of both the 3D scenes and their constituent objects.
>
> &emsp; This is a work worth exploring in the future: applying in-context learning to 3D scene-level point cloud segmentation to achieve the unification of semantic segmentation, instance segmentation, and panoptic segmentation. We would like to explore more possibilities of in-context learning in 3D point clouds, not only involving object-level point clouds.
>
> [Dai et al., 2017] Scannet: Richly-annotated 3d reconstructions of indoor scenes. In CVPR.
>
> [Chang et al., 2017] Matterport3d: Learning from rgb-d data in indoor environments. In 3DV.
>
> [Armeni et al., 2016] 3d semantic parsing of large-scale indoor spaces. In CVPR.
>
> >**Q3: What are the results of two combined prompts: denoising and part segmentation jointly?**
>
> **A3**:
>
> Caption: performance (mIOU) of PIC-Sep and PIC-Cat of part segmentation. The given prompts contain various levels of noise points ranging from 100 to 500 noisy points (1024 points per sample).
>
> | Models  | Original | Level=1 | Level=2 | Level=3 | Level=4 | Level=5 | Average |
> |:-------:|:--------:|:-------:|:-------:|:-------:|:-------:|:-------:|:-------:|
> | PIC-Sep |  74.95   |  75.13  |  75.02  |  74.94  |  74.84  |  74.68  |  74.92  |
> | PIC-Cat |  78.95   |  77.73  |  77.33  |  77.15  |  77.04  |  76.95  |  77.24  |
>
> &emsp; We use pre-trained PIC-Sep and PIC-Cat in Table 1 of the main text to test the combined task, when different levels of noises are added to the prompts during the testing. As the above table shows, **our PIC-Sep and PIC-Cat can naturally generalize to noised prompt without re-training**. For PIC-Sep, adding noise has little effect on its performance. For PIC-Cat, its mIOU drops about 1.2-1.7 in various levels of noise points. PIC shows great robustness. We **visualize** the part segmentation results of PIC-Sep and present the figure in **the PDF of the global rebuttal**.
>
> &emsp; Besides, we also notice that **the choice of prompt impacts the performance**. When we select the prompt for PIC according to the minimum Chamfer Distance between the query point cloud and the prompt (CD-aware), we find that the performance (mIOU) increases, as shown in the following table.
>
> Caption: The higher-quality prompt can improve the performance of PIC.
>
> | Models   | Random | CD-aware |
> |:---------:|:------:|:--------:|
> | PIC-Cat  |  78.95  |   **80.49** $\uparrow$ |
> | PIC-Sep  |  74.95  |   **78.46**$\uparrow$  |

---

> > ### Comment · Reviewer_Uvps · 2023-08-11
> > **Response to Author Rebuttal**
> >
> > Thanks for the response that addressed my concerns, and I will keep my positive score.

---

### Author Rebuttal · Authors · 2023-08-09

&emsp; We would like to thank the four reviewers for their suggestions, which make our paper more solid.

&emsp; We are grateful that the reviewers acknowledge our work. Here list some excerpts:

&emsp; **1.** As the reviewer **Uvps** and **GuqQ** said, Our work is the first to explore in-context learning for 3D point cloud understanding and is a very relevant but underexplored problem. Besides, the setup of in-context learning in 3D and the curation of the in-context learning dataset is helpful to the community.

&emsp; **2.** As the reviewer **uxmh** said, we unify the output space of the four tasks into the 3D coordinate space, and apply the MPM framework for model training, realizing one training, one model, and processing multiple tasks.

&emsp; **3.** As the reviewers **kyHc**, and **GuqQ** said, we design our Point-In-Context for in-context learning in 3D point clouds and address the "information leakage" problem via the Joint Sampling module. Such indicates that in-context learning in 3D is non-trivial.

&emsp; As suggested by four reviews, we conduct some additional experiments on our Point-In-Context and demonstrate more inspirable abilities.

&emsp; **1.** For the response of the reviewer **Uvps**, we use pre-trained PIC-Sep and PIC-Cat in Table 1 of the main text to test the combined task, when different levels of noises are added to the prompts during the testing part segmentation task. The results show our PIC-Sep and PIC-Cat **can naturally generalize to noised prompt without re-training**.

&emsp; **2.** For the response of the reviewer **GuqQ**, we conduct experiments on **other cross-modal SOTA MPM methods** like ACT, I2P-MAE, and ReCon. Such experiment results will be included in the final version.

&emsp; **3.** For the response of the reviewer **kyHc**, we conduct experiments that train our PIC for part of the tasks and test its emergent ability for the remaining tasks. Despite the formidable challenge, the PIC demonstrates **great emergent capability** on the other unseen task in the training.

&emsp; **4.** For the response of the reviewer **uxmh**, we explain more details about **task definitions, information leakage, and insights into our technological** methods.

&emsp; In conclusion, our work is **the first to explore in-context learning for 3D point cloud understanding**. We **unify the output space of the four tasks** into the 3D coordinate space, and apply the MPM framework for model training, realizing **one training, one model, and processing multiple tasks**. Extensive experiment results demonstrate the **superiority of the methodology design**, which also indicates that **in-context learning in 3D is non-trivial**. Besides, our Point-In-Context shows **great generalization capability**.

&emsp; We include two figures and two tables in **the PDF** and cite them in some rebuttal.

- Figure R1: Visualization of PIC-Sep on the part segmentation. The given prompts contain various levels of noise points ranging from 100 to 500 noisy points (1024 points per sample).

- Figure R2: The improved demonstration of information leakage in the previous MPM works.

- Table R1: The additional results of the cross-modal methods: ACT, I2P-MAE, and ReCon.

- Table R2: The numerical statistical results of target point clouds of the ShapeNet testing set and our reconstruction testing set.

---

### Decision · Program_Chairs · 2023-09-21

**Decision:**

Accept (spotlight)

**Comment:**

The paper delves into in-context learning for 3D point clouds, an important yet often overlooked issue. Reviewers unanimously commend the basic idea and concur on the method's efficacy towards its objectives. The authors have furnished a rebuttal addressing the reviewers' concerns. Consequently, the AC endorses the acceptance of the submission.